# Hidden Markov Model for Parkinson’s Disease Patients Using Balance Control Data

**DOI:** 10.3390/bioengineering11010088

**Published:** 2024-01-17

**Authors:** Khaled Safi, Wael Hosny Fouad Aly, Hassan Kanj, Tarek Khalifa, Mouna Ghedira, Emilie Hutin

**Affiliations:** 1Computer Science Department, Jinan University, Tripoli P.O. Box 818, Lebanon; 2College of Engineering and Technology, American University of the Middle East, Egaila 54200, Kuwait; hassan.kanj@aum.edu.kw (H.K.); tarek.khalifa@aum.edu.kw (T.K.); 3Laboratory Analysis and Restoration of Movement (ARM), Henri Mondor University Hospitals, Assistance Publique-Hôpitaux de Paris, 94000 Créteil, France; mouna.ghedira@aphp.fr (M.G.); emilie.hutin@aphp.fr (E.H.)

**Keywords:** HMM, machine learning, Parkinson’s disease, postural stability, stabilometric data

## Abstract

Understanding the behavior of the human postural system has become a very attractive topic for many researchers. This system plays a crucial role in maintaining balance during both stationary and moving states. Parkinson’s disease (PD) is a prevalent degenerative movement disorder that significantly impacts human stability, leading to falls and injuries. This research introduces an innovative approach that utilizes a hidden Markov model (HMM) to distinguish healthy individuals and those with PD. Interestingly, this methodology employs raw data obtained from stabilometric signals without any preprocessing. The dataset used for this study comprises 60 subjects divided into healthy and PD patients. Impressively, the proposed method achieves an accuracy rate of up to 98% in effectively differentiating healthy subjects from those with PD.

## 1. Introduction

The primary objective of the human postural system is to maintain stability in various static and dynamic situations, which involves responding to external perturbations during quiet standing and locomotion. This crucial function relies on different interactions among three main components: the first one is the central nervous system, the second one is the musculoskeletal system, and third one is the sensory systems, which consists of three parts, the vestibular, visual, and proprioception systems. Those interactions ensure the body remains upright [1,2,3,4,5].

It is obvious that Parkinson’s disease (PD) has a direct impact on the nervous system. PD is defined as a prevalent movement disorder that generates an impaired postural stability and increases the risk of physical injuries [6,7]. It causes abnormality of motor control in addition to degradation of the functionality of the basal ganglia rhythm generation, compromising the ability to maintain a controlled and upright posture in various situations and environments [8,9,10].

Numerous research studies have investigated PD in the context of both quiet standing and dynamic postures [11,12,13,14,15]. To distinguish between PD and non-PD subjects, data mining techniques have been employed using collected data from subjects during quiet standing or dynamic situations. Feature-extraction techniques can be employed to extract relevant information from collected data for the purpose of categorizing individuals as either having Parkinson’s disease (PD) or not [16,17]. The analysis of center-of-pressure (COP) displacements (COPDs) is a common method for assessing postural stability during quiet standing. These COPDs are typically registered in both orthogonal orientations: the medial–lateral (right/left) and anterior–posterior (forward/backward) directions. However, the conventional spatiotemporal analysis of COP output measures often is described, but the level of sensitivity is insufficient, and the method lacks the underlying control deficits. Specifically, quiet standing with eyes open exhibited a stronger interdependence between current and previous COPDs in comparison to experiments with closed eyes. However, there is no remarkable change in the vision conditions were observed for conventional parameters like the total length of the COP path or mean velocity. Experiment outputs highlighted the utility of AR parameters in evaluating postural stability for static posture with varying visual conditions.

Mei et al. [18] employed a support vector machine (SVM) to distinguish healthy and Parkinson’s disease patients using speech signals, achieving an accuracy of 85%. Barth et al. [19] utilized wearable sensors to monitor gait patterns and stages of the disease, achieving a 78% accuracy with a random forest classifier. These studies showcase the potential of machine learning techniques in Parkinson’s disease diagnosis. The main characteristic of Parkinson’s disease is the motor impairments, which include some disruptions in the balance control. Researchers in [20] examined the severity of the disease by analyzing postural sway data from force plates. They employed a decision tree classifier to classify patients into different severity levels, achieving an accuracy of 73%. This work highlights the importance of balance-related data in disease classification.

Hidden Markov models have been widely adopted for temporal pattern recognition tasks. Liu et al. [21] used HMMs to analyze handwriting patterns in Parkinson’s disease patients, achieving an accuracy of 91% in distinguishing patients from healthy individuals. This demonstrates the efficacy of HMMs in capturing subtle temporal variations in data for disease classification. In the subsequent section, we present the detailed methodology for our proposed hidden Markov model-based classification approach. We describe the data preprocessing steps, feature-extraction techniques, and the formulation of the hidden Markov model to facilitate the uses of balance control data to classify Parkinson’s disease patients.

Authors in [22] tested some samples of PD subjects with accelerometer-based data to study postural stability during the quiet stance. The study involved computing 175 measures for different frequencies and time values, then feature selection was carried out based on a classification model in order to identify discriminative parameters to differentiate control and PD subjects. This differentiation is finally based on two parameters. It is worth noting that the process of feature extraction, classification, filtering, selection, and training may introduce extra computational overhead that could pose challenges for real-time analysis.

In previous work [23], authors proposed a novel model for assessing postural stability by analyzing center-of-pressure (COP) trajectories for quiet standing scenarios. To investigate all probable changes in the postural stability which are related to the visual input, they extracted additional sensitive parameters. The study involved 11 healthy subjects belonging to the age interval [20, 27]. They used a force platform to record their stabilometric signals in both eyes-open and eyes-closed conditions during quiet standing. The new model was separately tested with two different types of stabilometric signals: medial–lateral (ML) and anterior–posterior (AP).

In [24], the authors studied the application of the AR model to analyze stabilometric signals related to different subjects and conditions. This study was established separately for ML and AP conditions, where M = 20 (the order of the AR models). Those novel measurements obtained from the estimation of the AR model parameters highlight the percentage contributions and geometrical moment of AR coefficients.

They exhibited acceptable remarkable distinction of eye conditions (open and closed). Notably, for an eyes-open scenario in the quiet standing condition, results show an important interdependence between current and previous COP displacements in comparison with the eyes-closed condition. However, for classical parameters (such as the overall length of the COP path and the mean velocity), the difference between vision conditions was minor and sometimes could be neglected. The outputs indicated that the evaluation of postural stability using AR parameters with consideration of different visual conditions is more important and relevant than that considering the same visual conditions.

In contrast to the conventional approach based on COP characteristics, Blaszczyk [25] used force-plate posturography to check postural stability. The study involved 168 subjects categorized as healthy and PD patients and classified into two groups: young and elderly. Subjects were tested in two different scenarios: both eyes open and then with both eyes closed. Authors in this work focused on measuring and analyzing three different outputs: sway ratio (SR), sway directional index (DI), and sway vector (SV). Those outputs are directly impacted by many parameters such as age, pathology, and visual conditions, and show important differences among different groups. The sway vector was selected as the most adequate variable for evaluating postural control during quiet standing scenarios.

In our previous paper [26], the empirical mode decomposition (EMD) method was used to distinguish healthy and PD patients using the same dataset of the present study. EMD allows the decomposition of a complex signal into many simpler signals called intrinsic mode functions (IMFs). As a final result for this previous study, the random forest method had a performance of 94%, and the Dempster–Sahfer formalism method had a very high accuracy of 96.51%.

In a neurological clinic specializing in Parkinson’s disease (PD), a significant impact can be observed through various areas. (1) Diagnostic advancements: Improved diagnostic tools and techniques can lead to earlier and more accurate detection of PD. This could involve advanced imaging technologies or biomarker identification, enabling timely intervention and better disease management. (2) Treatment innovations: Developing new therapies or refining existing ones can enhance symptom management and slow disease progression. This might involve novel medications, deep brain stimulation techniques, or other innovative treatments tailored to address specific symptoms or stages of PD. (3) Personalized medicine: Tailoring treatment plans according to individual patient profiles, genetics, and disease progression can significantly improve outcomes. Customized approaches can optimize medication regimens and therapeutic interventions for each patient’s unique needs. (4) Patient care and support: Implementing comprehensive care models that focus on holistic support for PD patients and their families is crucial. This includes multidisciplinary care teams, support groups, counseling, and rehabilitation programs aimed at improving quality of life and managing non-motor symptoms. (5) Research and education: Continued research efforts into understanding the underlying mechanisms of PD can lead to breakthroughs in treatment and care. Education programs for both patients and healthcare providers can ensure the latest advancements are disseminated and implemented effectively. (6) Technological interventions: Utilizing technology such as wearable devices for monitoring symptoms, telemedicine for remote consultations, and digital platforms for data collection can enhance monitoring and care delivery, especially in remote or underserved areas. Advancements in diagnostics, treatment modalities, personalized care, patient support, ongoing research, and technological integration can collectively bring about a significant positive impact in a neurological clinic specializing in Parkinson’s disease. These advancements aim to improve patient outcomes, enhance quality of life, and ultimately work towards finding a cure for PD.

The present study presents a classification approach using hidden Markov models (HMMs) to differentiate healthy subjects from those with PD. Notably, this approach directly utilizes raw stabilometric signals without any feature-extraction and feature-selection steps. The HMMs were constructed based on medial–lateral (ML), anterior–posterior (AP), or combined ML and AP signals. The 10-fold cross-validation method was applied to divide the dataset into training and testing datasets. The list of contributions can be represented as follows:The paper introduces an innovative approach that utilizes hidden Markov model (HMM) to distinguish between healthy individuals and those with PD.The proposed approach employs raw data obtained from stabilometric signals without any preprocessing.The proposed model obtains a very high accuracy rate that can reach up to 98% while effectively differentiating healthy subjects from those with PD.The classification performance obtained using the proposed approach is superior to that reported in previous studies.The proposed approach holds promise for various posture analysis applications including identifying the severity of pathology in patients and predicting falls.

The rest of this paper is organized as follows: Section 2 presents the hidden Markov models (HMMs) applied in this work. Section 3 describes the data acquisition process and explains the methodology used in the classification process of healthy and PD subjects. Section 4 discusses the performance results generated by the application of the HMM. Section 5 gives the conclusion of the paper and describes the probable extension of the work.

## 2. Hidden Markov Models

Initially introduced in [27] between 1965 and 1970, a hidden Markov model (HMM) is a statistical framework characterized by a configuration comprising various states connected by transitions. This model shares similarities with probabilistic automata but has a crucial difference: symbol generation occurs at the states rather than during transitions. Furthermore, in contrast to automata, each state in an HMM is associated with probability distributions for all symbols in the alphabet. HMMs are powerful tools for analyzing temporal or sequential data and have applications in various domains, including DNA and RNA sequencing, voice recognition, handwriting recognition, activity recognition, and more.

### 2.1. State of the Art

The utilization of hidden Markov models (HMMs) in analyzing balance control data for Parkinson’s Disease (PD) patients has emerged as a promising avenue in the field of healthcare technology. HMMs are a type of probabilistic model that have applications in various fields due to their ability to model temporal sequences and capture hidden states governing observed data [16,28]. The state of the art in using an HMM for Parkinson’s disease patients using balance control data is as follows. Data collection and preprocessing: Researchers collect balance control data using various wearable sensors (accelerometers, gyroscopes) or force platforms. These devices capture movement patterns, gait dynamics [29], sway, and other balance-related parameters [30]. Preprocessing involves noise removal, filtering, and feature extraction to obtain meaningful information from the raw sensor data [31]. Relevant features extracted from balance control data include sway velocity, sway area, sway amplitude, gait characteristics, center of pressure displacement, and spectral analysis parameters. These features serve as inputs for HMMs. Model development with HMMs: The HMMs are employed to model the temporal dynamics of the extracted features [32]. The model typically consists of observable variables (features extracted from balance control data) and hidden states (representing underlying movement patterns or stages related to PD). The hidden states in the HMM may correspond to different stages of Parkinson’s disease or distinct motor states (e.g., tremor-dominant, gait-dominant, or dyskinesia) [9]. The transition probabilities between these states capture the temporal dynamics of the disease progression or motor state changes. Estimation techniques such as the expectation–maximization (EM) algorithm or Baum–Welch algorithm are employed to estimate the model parameters, including transition probabilities, emission probabilities, and initial state probabilities. Trained HMMs can be used for classification tasks to distinguish between PD and healthy control subjects based on their balance control data. Additionally, HMMs can predict the progression of PD stages or forecast the onset of specific motor symptoms [33]. Clinical applications [32]: The application of HMMs in analyzing balance control data for PD patients has significant clinical implications. It can aid in early diagnosis, disease monitoring, personalized treatment planning, and assessing the effectiveness of interventions (medications or physical therapy). Challenges include the need for large, diverse datasets; interpretability of hidden states; model generalizability across different patient populations; and the integration of multi-modal data sources for a comprehensive understanding of PD progression. Future research might focus on incorporating deep learning techniques for enhanced feature representation and the development of hybrid models combining HMMs with other machine learning approaches. The use of hidden Markov models in analyzing balance control data for Parkinson’s Disease patients holds promise in understanding disease progression, assisting in clinical decision-making, and developing personalized interventions for better patient care. Further advancements in model sophistication and data collection techniques are essential for maximizing the utility of this approach in clinical practice.

### 2.2. Markov Chain

Prior to exploring the HMM, it is crucial to introduce the probabilistic model for observable sequences, known as the observable Markov model or Markov chain.

A Markov chain (MC) [34] serves as a stochastic process employed to model diverse sequential and temporal phenomena across numerous application domains. The initial distribution of states and the probabilities governing transitions between these states are the main factors used to define the observable sequence in a Markov chain. Typically, a Markov chain generates many elements within the same observable sequence. Those elements are expected to exhibit temporal dependence.

For a first-order MC, the probability of changing the present state depends on the old state sequence, which depends only on the old state. Formally:(1)$$p\left(z_{n}|z_{n-1},z_{n-2},\ldots ,z_{1}\right)=p\left(z_{n}|z_{n-1}\right)$$
where $z_{n}$ represents a series of random variables, applicable for all values of *n* greater than zero.

A *p*-order MC is defined as a set of random variables that meet the conditions mentioned in Equation (Equation 2):(2)$$p\left(z_{n}|z_{n-1},z_{n-2},\ldots ,z_{1}\right)=p\left(z_{n}|z_{n-1},z_{n-2},\ldots ,z_{n-p}\right)$$
where $z_{n}$ consists of a set of random variables, for all *n* larger than both zero and *p*.

A classical MM can be defined by:*K*: the count of model states, similar to the size of the employed alphabet in terms of the number of characters.$A=a_{kl}$: the transition probability matrix, denoted as $a_{kl}=p\left(s_{l}|s_{k}\right)$, where 1<k,l<K and $\sum_{l=1}^{K}a_{kl}=1$. This matrix represents the probabilities of transitioning from state $s_{k}$ to state $s_{l}$.π: the initial probability vector, denoted as $\pi_{k}=p\left(s_{k}\right)$, signifies the initial probability associated with state $s_{k}$, where 1<k<K.

### 2.3. Discrete HMM

The HMM belongs to the class of statistical models. These models integrate many stochastic approaches, which are divided into two processes. A Markov chain detailing the hidden state sequence is used to illustrate the first process. A sequence of random parameters that describe the observation series is used to define the second process.

Generally, the description of an HMM is slightly different from the representation of an observable Markov model, as the former is described through indirect illustration/representation of the states. An HMM generates statements generally weighted by their probabilities. Consider $S=s_{1},s_{2},\ldots ,s_{K}$ denotes a series of states and $O=o_{1},o_{2},\ldots ,o_{M}$ represents all characters in the observable alphabet. Here, *K* counts the states, and *M* is the characters. Consider a first-order HMM, and assume $X=x_{1},x_{2},\ldots ,x_{n}$ is the series of statements produced during the application of the HMM through HM chain series $Z=z_{1},z_{2},\ldots ,z_{n}$. It is vital to point out that every statement $x_{i}$ is linked to one value from *O*, and the HMM $z_{i}$ takes its value from the set *S* for i between 1 and n.

An HMM is defined as:*K*: the whole count of the model states;*M*: equal to the number of all characters in the set *O*;$A=a_{kl}$: the matrix of the probability transition, with $a_{kl}=p\left(z_{i}=s_{l}|z_{i-1}=s_{k}\right)$, that stands for the likelihood of transition having as a source $s_{k}$ and destination $s_{l}$, where k is larger than 1 and K is larger than l and $\sum_{l=1}^{K}a_{kl}=1$;The likelihood of emitting character $o_{m}$ from state $s_{l}$ is represented as: $p(x_{i}=o_{m}|z_{i}=s_{l})$. The likelihood of all probable emission scenarios are shown in the matrix of emission likelihoods $B=b_{lm}$, where $b_{lm}=p\left(x_{i}=o_{m}|z_{i}=s_{l}\right)$ for m between 1 and M and l between 1 and kπ: the set of of earliest likelihoods, denoted as $\pi_{k}=p\left(z_{i}=s_{k}\right)$, represents the early likelihood of state *k* for any value of k between 1 and K.

Figure 1 presents a discrete HMM where every character $x_{i}$, linked to an invisible state $z_{i}$, is defined by its emission likelihood.

When employing an HMM for time series modeling, many key considerations have to be taken into account.
The evaluation problem: Considering the HMM λ and the series of observations, what is the likelihood that the prescribed model can generate such observations? The objective is to count the likelihood p(X|λ) to produce the sequence of observations $X=x_{1},x_{2},\ldots ,x_{n}$ through the HMM λ=π,A,B. This likelihood is equivalent to the total of all likelihoods that *X* may produce through all probable state series. In general, the forward–backward algorithm is applied to overcome this issue [34].The decoding problem: With the HMM λ and the series of observations *X*, how can the optimum series of states (hidden sequence) that generates the observations sequence be determined? In other words, what is the most probable state series of the model λ that produces the observations sequence *X*? The Viterbi algorithm is mainly applied to overcome this issue [35].The learning problem involves adjusting the inputs of the mode λ=π,A,B to increase the likelihood of the observations series p(X|λ). The expectation–maximization (EM) algorithm is frequently employed for learning the HMM.


**Parameter Estimation:**


In the context of a first-order hidden Markov chain, the state sequence distribution $Z=(z_{1},z_{2},\ldots ,z_{n})$ can be expressed as mentioned in Equation (Equation 3):(3)$$p\left(Z;\pi ,A\right)=p\left(z_{1};\pi \right)\prod_{i=2}^{n}p\left(z_{i}|z_{i-1};A\right)$$

The conditional distribution of the observations series *Y* considering the series of state *Z* can thus be represented as shown in Equation (Equation 4):(4)$$p\left(X|Z;\lambda \right)=\prod_{i=1}^{n}p\left(x_{i}|z_{i};\lambda \right)$$

In conclusion, the joint distribution of *X* and *Z* (referred to as the complete-data likelihood) can be formulated as illustrated in Equation (Equation 5):(5)$$\begin{array}{c} p(X,Z;\lambda )=p(Z;\pi ,A)p(X|Z;A)\\ =p\left(z_{1};\pi \right)p\left(x_{1}|z_{1};\lambda \right)\prod_{i=2}^{n}p\left(z_{i}|z_{i-1};A\right)p\left(x_{i}|z_{i};\lambda \right) \end{array}$$

For an HMM described by λ=π,A,B, some parameters need to be estimated such as the first distribution π, the matrix containing all transition likelihoods *A*, and the matrix of emission probabilities *B*. The highest probability method is employed to optimize the log-likelihood of the observation data. However, analytically maximizing this log-likelihood proves to be highly challenging. The specialized expectation–maximization (EM) algorithm for HMMs, often referred as the Baum–Welch algorithm, is frequently utilized for this purpose [27].

### 2.4. Gaussian HMM

In many implementations, the observations are treated as analog values. In these instances, the emission likelihood of every state is typically modeled using the density function of a Gaussian model. Consequently, the conditional state density (the likelihood of emission) is expressed in:(6)$$p\left(x_{i}|z_{k};\lambda \right)=N\left(y_{i};\mu_{k},\Sigma_{k}\right),$$
with:N representing the function of the Gaussian probability density;$\mu_{k}$ presenting the average of the observation distribution at state *k* for all k=1,2,…,K;$\Sigma_{k}$ standing for the matrix of variance-covariance for state *k* for all k in the set 1,2,…,K.

Figure 2 illustrates a sample of a continuous HMM with Gaussian emission probabilities.

To analyze and evaluate the inputs in any Gaussian HMM, the equations for updating the first likelihood π and the moving likelihood in a discrete HMM remain unchanged. However, for the probability of emission, the Gaussian distribution inputs $\mu_{k}$ and $\Sigma_{k}$ need to be computed for every state. This is accomplished using the following:(7)$$\mu_{k}^{\left(q+1\right)}=\frac{1}{\sum_{i=1}^{n}\tau_{ik}^{\left(q\right)}}\sum_{i=1}^{n}\tau_{ik}^{\left(q\right)}x_{i}$$
(8)$$\Sigma_{k}^{\left(q+1\right)}=\frac{1}{\sum_{i=1}^{n}\tau_{ik}^{\left(q\right)}}\sum_{i=1}^{n}\tau_{ik}^{\left(q\right)}\left(x_{i}-\mu_{k}^{\left(q+1\right)}\right){\left(x_{i}-\mu_{k}^{\left(q+1\right)}\right)}^{T}$$

Here, $\mu_{k}^{\left(q+1\right)}$ and $\Sigma_{k}^{\left(q+1\right)}$ represent the updated values of the mean and covariance matrix of state *k* at iteration (q+1), respectively. The parameters $\tau_{ik}^{\left(q\right)}$ are the posterior probabilities obtained from the forward–backward algorithm and used as weights in the update process. The superscript (q) indicates the iteration step of the parameter estimation process.

## 3. Methodology

We believe that mathematical studies play a vital role in understanding Parkinson’s disease (PD) from various angles. Modeling disease progression: Mathematical models can simulate the progression of PD, allowing researchers to understand how the disease develops and spreads in the brain. These models often incorporate factors such as neuronal degeneration, neurotransmitter dynamics, and motor symptom progression. Biomechanical analysis: Mathematics is used to analyze movement patterns and motor control in PD patients. Computational models and mathematical algorithms help in understanding gait abnormalities, tremors, and other motor symptoms by quantifying and characterizing these movements. Network analysis of brain connectivity: Graph theory and network analysis are employed to study the brain’s connectivity patterns and how disruptions in neural networks might contribute to PD. These studies help in understanding the underlying changes in brain connectivity associated with PD symptoms. Pharmacokinetics and drug development: Mathematical models are used to simulate drug interactions, pharmacokinetics, and drug dynamics in the body. These models aid in predicting drug effects, optimizing dosages, and developing new therapies for PD. Epidemiological studies: Mathematics is instrumental in epidemiological studies related to PD, including analyzing risk factors, disease prevalence, and genetic influences. Statistical modeling helps in identifying trends and risk associations within populations. Optimization of deep brain stimulation (DBS): Mathematical algorithms assist in optimizing the placement of electrodes and parameters for DBS, a surgical treatment for PD. These algorithms aim to maximize therapeutic effects while minimizing side effects. Data analysis and computational techniques: Mathematics is fundamental in analyzing large datasets generated from clinical studies, genetic studies, brain imaging, and other sources. Statistical methods and machine learning algorithms help identify patterns, correlations, and predictive models related to PD. Clinical trial design: Mathematical modeling contributes to the design and optimization of clinical trials, aiding in sample size determination, treatment evaluation, and outcome prediction. In summary, mathematical studies provide a quantitative and analytical framework for understanding various aspects of Parkinson’s disease. These studies contribute significantly to unraveling the disease’s complexities, improving treatment strategies, and potentially finding better therapeutic interventions. As a future development of this work, we will definitely contribute towards the clinical study.

This section outlines the steps followed to acquire data and describe the proposed methodology applied for sampling the healthy subjects and subjects with Parkinson’s disease.

### 3.1. Data Acquisition

The data used in this study were acquired at the well-known hospital in Creteil, France (Mondor Hospital), the location of experiments. The dataset comprises stabilometric signals obtained from healthy persons and PD patients. A total number of 60 subjects were added to the dataset (28 healthy subjects and the remaining PD patients). Samples were divided into young and old adults (12 young PD patients and 20 older adults). Table 1 provides main criteria/specifications regarding the PD selected subjects.

Our study is a preliminary study, and we are planning to validate the mathematical model on various H&Y score groups. Patients were undergoing a drug holiday prior to testing. We are also planning to validate the mathematical model on different age groups. The duration of a drug holiday for Parkinson’s disease can vary widely based on individual circumstances and the specific medications being taken. A drug holiday refers to a deliberate interruption or reduction in the dosage of medication to manage Parkinson’s symptoms. This strategy aims to address potential medication-related complications, such as dyskinesia, that can develop with long-term use of certain Parkinson’s medications, particularly levodopa. Drug holidays are typically managed under the guidance and supervision of a healthcare professional, often a neurologist specializing in movement disorders. The length of a drug holiday can be tailored to the individual’s needs, balancing the management of symptoms with minimizing side effects. These holidays can range from a few days to several weeks, depending on the specific circumstances and the response of the person with Parkinson’s disease to the medication changes. In our study, a period of six days was taken on average as the duration of the drug holiday for the patients.

The Hoehn and Yahr (H&Y) scale is a commonly used system for assessing the progression of Parkinson’s disease. It is used by clinicians to determine the severity of symptoms and functional disability in individuals with Parkinson’s. The scale ranges from stages 1 to 5, with higher stages indicating more severe impairment and disability. Stage 1: This stage represents the mildest form of Parkinson’s disease. Typically, at this stage, symptoms are unilateral (affecting one side of the body) and may not significantly interfere with daily activities. Tremors and other motor symptoms might be present but are mild. Stage 2: In this stage, both sides of the body are affected, but balance is usually not severely impaired. Daily activities can still be performed without substantial disability. Stage 3: This stage signifies moderate disability. Balance issues become more prominent, and individuals might experience a greater risk of falling. However, they can still function independently with some difficulty. Stage 4: At this stage, symptoms are severe, and individuals often require assistance with daily activities. They may still be able to walk but with considerable difficulty and typically need assistance. Stage 5: This is the most advanced stage of Parkinson’s disease. Individuals at this stage are usually wheelchair-bound or bedridden. They may require round-the-clock care due to the severity of their symptoms and functional limitations. The selection of the appropriate H&Y score for a person with Parkinson’s disease is based on a clinical assessment by a healthcare professional. Factors such as motor symptoms, functional abilities, balance, and overall impact on daily life are considered to determine the most appropriate stage on the scale. Our study is a preliminary study where testing is conducted at stage 2. Other stages will be tested as in a future development of this study. In the future work, machine learning models will be used to classify subjects based on their PD stages.

While recording the stabilometric signals, all subjects were asked to perform quiet standing in the anterior–posterior (AP) and medial–lateral (ML) directions for 1 min. AP and ML trajectories were used to represent the center of pressure movements in the forward/backward and the right/left directions of the human body, respectively.

The measurements were carried out using a 6-component force plate (60 × 40 cm, strain gauge based device from Bertec Corporation, Columbus, OH, USA) with a sampling rate of 1000 Hz.

### 3.2. HMM-Based Classification Approach

This subsection introduces the HMM-based approach designed for distinguishing between healthy and Parkinson’s disease (PD) subjects based on their Center of Pressure (COP) displacements during quiet standing. The methodology entailed developing two Gaussian HMMs: the first model (H-HMM) parameters were learned from the training dataset of healthy subjects, whereas the second model (PD-HMM) parameters were learned from the training dataset of PD subjects. The implementation of every HMM is founded on the serial structure of the training signals.

The classification of the subjects as healthy/PD was accomplished following two iterations: the first consists of using either ML or AP stabilometric signals, and the second is represented by the incorporation of signals derived by ML and AP directions.

The classification process for test subjects was as outlined below: For every test subject, the procedure illustrated in Figure 3 was executed in the following manner: the probability of observations for the test subject was calculated for the H-HMM and PD-HMM models. The class to which the test subject belongs was determined by selecting the highest probability between these two models. The test subject was categorized as healthy if the H-HMM yielded the largest probability; otherwise, the test subject was classified as having Parkinson’s disease (PD).

## 4. Results and Discussion

In this section, experimental results and comparison with other studies are provided.

### 4.1. Experimental Results

This section discusses the classification performance obtained using the HMMs. As mentioned before, an HMM is defined by many inputs, including the first likelihood, the transition likelihood matrix, and the Gaussian model of every state. The number of states and Gaussian mixtures considered in the model can also influence its performance. Additionally, the number of iterations in the log-likelihood optimization process during model training is a critical parameter that can affect the model’s effectiveness.

In this investigation, the K-means algorithm was employed to predict the first values of Gaussian model parameters, including the mean $\mu_{k}$ and the covariance matrix $\Sigma_{k}$ for every state $s_{k}$. Subsequently, the Baum–Welch algorithm was applied to refine these parameters by optimizing the log-likelihood of the training dataset. Figure 4 illustrates the progression of the log-likelihood probability during the maximization process for AP, ML, and AP/ML signals. It is evident that the log-likelihood outputs coincide after the 15th iteration for both the H-HMM and PD-HMM. Hence, the count of steps for log-likelihood optimization is equal to fifteen.

To ascertain the most suitable number of states in the HMMs, we analyzed their performance at different state numbers. Table 2 shows the accurate classification rates for healthy and Parkinson’s disease (PD) subjects employing the two HMMs with new values of states.

It is evident that all achieved accuracy rates are 90% or higher, and the optimal classification performance is attained when employing three states.

Similarly, we examined the changes in the HMM performance as a function of the number of Gaussian mixtures. Table 3 presents the accurate accuracy rates for healthy and Parkinson’s disease (PD) patients using the two HMMs with varying numbers of Gaussian mixtures.

As the number of parameters increases in the HMM, the model becomes more complex, and hence the accuracy drops. This leads to a significant degradation of the prediction that the HMM performs (Table 2). The same observations were made for the number of Gaussian mixtures; within each state we used a specific number of Gaussian mixtures, and we found that with two Gaussian mixtures inside each state we can obtain a better result, but if we instead increase the number of Gaussian mixtures within each state, results will be degraded, and prediction and decision of the model will be deteriorated (Table 3).

In addition, the change in the HMM performance based on the number of Gaussian mixtures was analyzed. All accuracy rates were above than 89%, and the optimal performance was achieved with two Gaussian mixtures. In conclusion, the optimal performance was obtained when setting the number of iterations for log-likelihood maximization to 15, the number of states to 3, and the number of Gaussian mixtures to 2.

As the number of parameters increases, the computational complexity of training the model also increases. This can result in longer training times and higher resource requirements.

The evaluation of HMMs was conducted using the 10-fold cross-validation method. Three key statistical parameters, namely sensitivity, specificity, and overall accuracy, were calculated to assess the performance of the HMMs:-Sensitivity signifies the percentage of accurately classified healthy subjects;-Specificity corresponds to the percentage of correctly classified subjects with Parkinson’s disease (PD);-Accuracy denotes the percentage of both healthy and Parkinson’s disease (PD) subjects who are accurately classified;

Table 4 presents the classification performance of the HMMs. Only one PD subject was misclassified, representing a sensitivity of 100% and a specificity of 96.4%. The total accuracy for classifying healthy and PD subjects was 98.4%. This table also displays the performance of the new classification approach when utilizing only AP signals or ML signals. The performance achieved using ML signals was comparable to that obtained using both ML and AP signals, yielding an overall accuracy of 98.4%. When utilizing AP signals, all subjects were classified correctly except for one healthy subject and one PD subject, resulting in an overall accuracy of 96.6%.

Figure 4 illustrates the log-likelihood maximization process of the HMMs for healthy and PD subjects using both AP and ML signals.

### 4.2. Comparison with Other Studies

In our previous paper, the same dataset was used to distinguish healthy and PD subjects. After applying the EMD method over the stabilometric signals, temporal and spectral features were extracted from each stabilometric signal as well as from its IMFs. Many well-known machine learning methods were used, namely the KNN, decision tree, random forest and SVM classifiers. The random forest method had an accuracy of 94%, and the Dempster–Sahfer formalism method had an accuracy of 96.51%. In contrast, in this present study, no feature extraction or feature selection techniques were used, and we achieved a better accuracy rate (98%) than the best accuracy achieved in the previous paper (96.51%).

Another study [36] involved feature selection and classification processes. The authors utilized feature importance and recursive feature elimination methods for feature selection and employed classification and regression trees, artificial neural networks, and support vector machines for classification. Their achieved accuracy was 93.84%. In contrast, our proposed method achieved a higher accuracy level of 98%.

## 5. Conclusions and Future Work

This work introduces a novel strategy for effectively classifying healthy and PD patients. The final results demonstrate that our proposed model can achieve accurate classification rates of up to 98% for distinguishing between healthy and PD subjects based on HMMs. Moreover, the classification performance resulting from the implementation of our approach was superior to that reported in previous studies. By utilizing ML and AP signals, it was clearly seen that all subjects were correctly classified except one PD subject. The sensitivity was equal to 100%, and the specificity was 96.4%, with a total accuracy of 98.4%.

The new approach holds promise for various posture analysis applications, including identifying the severity of pathology in patients and predicting falls.

As future work, the proposed methodology can be applied to a wider dataset to validate and verify its effectiveness. A larger number of samples/subjects should encompass PD subjects of various age groups and different stages of PD to ensure a representative sample of the PD population. One possible extension to the current work is using deep learning techniques such as recurrent neural networks (RNNs) as a possible extension to the current study. Using RNNs to enhance Parkinson’s disease prediction offers several advantages:Sequence modeling: RNNs are well-suited for handling sequential data, which is crucial in analyzing time series data. Symptoms and patient data often occur in a temporal sequence, and RNNs can capture these temporal dependencies effectively.Feature extraction: Deep learning models, including RNNs, can automatically learn relevant features from raw data. This involves the time series measurements.Complex pattern recognition: RNNs can capture intricate patterns and correlations in the data that might not be easily discernible using traditional machine learning techniques. This ability can help in identifying subtle changes in disease progression or predicting the onset of symptoms.

## Figures and Tables

**Figure 1 bioengineering-11-00088-f001:**
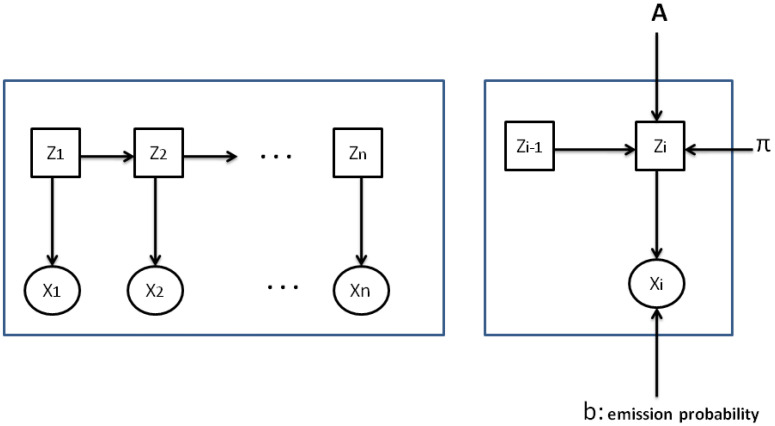
An example of a discrete hidden Markov model.

**Figure 2 bioengineering-11-00088-f002:**
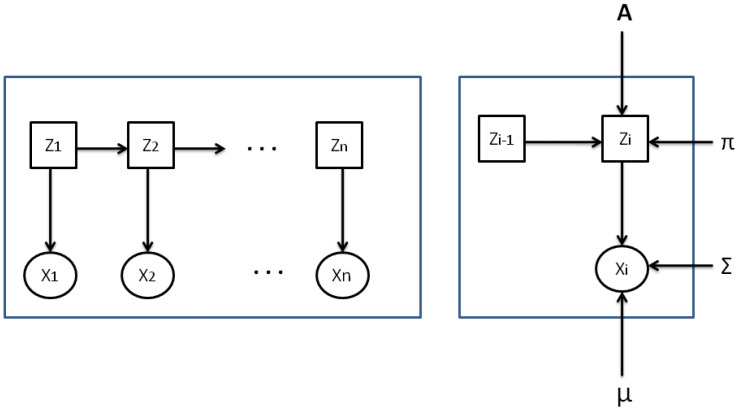
A sample of a Gaussian HMM.

**Figure 3 bioengineering-11-00088-f003:**
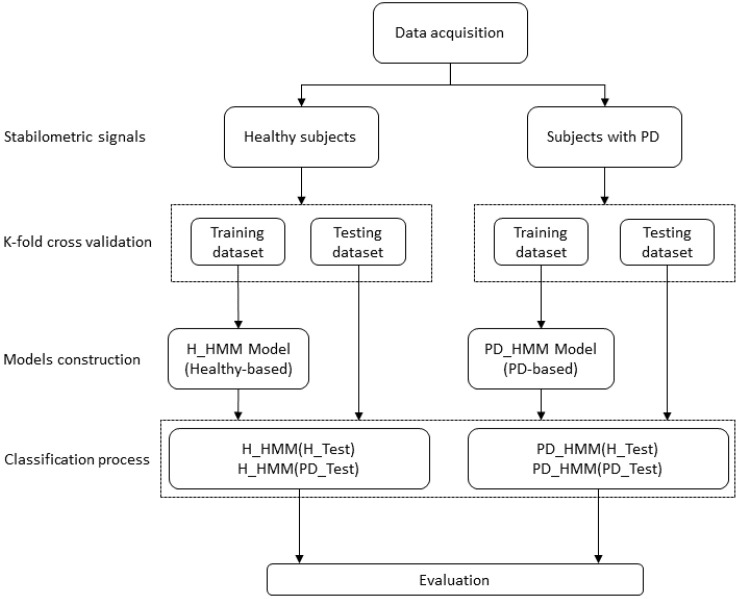
Proposed classification approach diagram.

**Figure 4 bioengineering-11-00088-f004:**
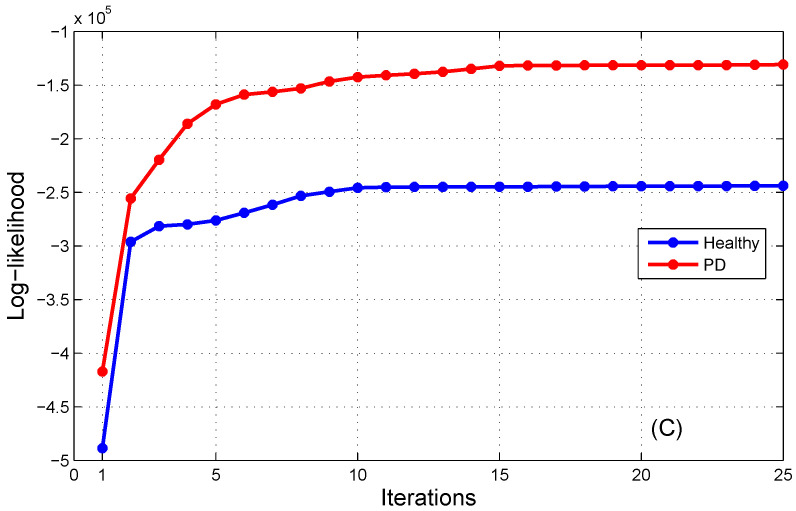
The log-likelihood maximization process of the HMMs using both AP and ML signals.

**Table 1 bioengineering-11-00088-t001:** Description of the PD population.

Criteria	Value
Age (mean ± SD)	67 ± 8 years
Time since diagnosis	8 ± 5 years
Score (Hoenh and Yahr)	2.2 ± 0.3
Weight	75 ± 18 kg
Height	167 ± 11 cm

**Table 2 bioengineering-11-00088-t002:** Classification rates with different counts of states.

**States number**	2	3	4	5	6	7	8
**Accuracy rates**	93.3%	98.4%	96.6%	93.3%	95.0%	90.0%	90.0%

**Table 3 bioengineering-11-00088-t003:** Classification rates with different numbers of Gaussian mixtures.

**Number of Gaussian mixtures**	2	3	4	5	6	7	8
**Accuracy rates**	98.4%	97%	97%	95%	95%	90%	91.6%

**Table 4 bioengineering-11-00088-t004:** Performance of the proposed classification approach for differentiating between healthy and PD subjects (1) using both ML and AP signals, (2) using ML signals, (3) using AP signals.

(1)	**Subjects**	**Predicted H**	**Predicted PD**	**Sensitivity/Specificity**	**Overall Accuracy**
**Healthy**	28	28	0	100%	98.4% ± 0.3%
**PD**	32	1	31	96.8%	
(2)	**Subjects**	**Predicted H**	**Predicted PD**	**Sensitivity/Specificity**	**Overall Accuracy**
**Healthy**	28.0	27.0	1.0	96.4%	96.6% ± 0.3%
**PD**	32.0	1.0	31.0	96.8%	
(3)	**Subjects**	**Predicted H**	**Predicted PD**	**Sensitivity/Specificity**	**Overall Accuracy**
**Healthy**	28.0	28.0	0.0	100.0%	98.4% ± 0.3%
**PD**	32.0	1.0	31.0	96.8%	

## Data Availability

Experimental Data was provided by the ARM Laboratory ARM, CHU Henri Mondor, Créteil France.

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
