# Peer review of "Hidden Markov Model for Parkinson’s Disease Patients Using Balance Control Data"

_bioengineering, 2024, doi:10.3390/bioengineering11010088_

Round 1
Reviewer 1 Report (Previous Reviewer 3)
Comments and Suggestions for Authors
authors answered most questions although lacking perfectness on Q3 and Q4.
Regarding Q4: authors did not provide how long the drug holiday lasted and what changes were observed on patients after the durg holidays.
Author Response
Please find the feedback attached

Reviewer 2 Report (Previous Reviewer 2)
Comments and Suggestions for Authors
This paper proposes the use of stabilometric signals for detecting PD using HMMs. The number of participants is very high, and the analyses are interesting. I think the paper has improve significantly.
Minor comments to improve the paper:
- Improve the quality of figure 3.
- Include an introductory paragraph in section 4 (before subsection 4.1)
- In table 4, I think the confidence intervals are not correct. The idea es to say 08% of accuracy +- 2% with a 95% confidence. I miss the possible variation
- I’d suggest expanding the discussion regarding tables 2 and 3. Why we obtain worse results when increasing the number of parameters?
Author Response
Please find the reply attached

This manuscript is a resubmission of an earlier submission. The following is a list of the peer review reports and author responses from that submission.
Round 1
Reviewer 1 Report
Comments and Suggestions for Authors
This research introduces an innovative approach using the Hidden Markov Model to distinguish between healthy individuals and those with Parkinson's disease (PD). The method uses raw data from stabilometric signals without preprocessing, achieving an accuracy rate of up to 98% in classifying healthy subjects from PD patients. The dataset is very small to reach to come to a conclusion. Authors may improve the number of the dataset or change the title.
Comments on the Quality of English LanguageIt is fine.
Reviewer 2 Report
Comments and Suggestions for Authors
This paper proposes and evaluates the use of HMM for Parkinson Disease detection using stabilometric signals. The paper is well designed and presented, but I think the novelty is very low. Perhaps, the use of this type of signals can be a novelty. In this case, I’d suggest including a more detailed description of this type of signals.
Comments to improve the paper:
- At the end of the introduction include a list of contributions.
- I’d suggest including a state of the art section
- Section 2, in a section before a subsection title, I’d suggest an introductory paragraph
- Regarding the dataset. How many recordings per participants? When splitting the data into training and testing, are you considering recordings from the same subject in training and testing?
- When including figures in test or tables, I’d suggest including the same number of decimals.
- In the results tables, I’d suggest including confidence intervals to see the significance of the results.
- Have you tried using deep learning, RNNs for example?
- I’d suggest expanding section 4.2 including more previous studies.
Reviewer 3 Report
Comments and Suggestions for Authors
Parkinson’s disease (PD) is the second most common, progressive neurodegenerative disorder in elderly and the prevalence of which is on the rise. The diagnosis and management of PD is therefore likely to become increasingly frequent in general practice. Moreover, early diagnosis of PD is often complicated by other complications with similar clinical features such as normal pressure hydrocephalus, progressive supranuclear palsy and multiple system atrophy, which become more challenging to make correct and early diagnosis of the disease (Waller et al., 2021). It would be help if the methods described in the present manuscript is proven practically useful for physicians.
A lack of important information is a major obstacle to be convinced the method has significant impact in neurological clinic
· The study is a mathematic- rather than clinical-oriented and has very limited medical information provided. In the study, authors tried to use a classification approach using Hidden Markov Models (HMM) to differentiate healthy subjects from those with PD. However, it may be difficult to predict its clinical significance or potential due to lack of clinically relevant discussions in the current version of manuscript.
· "PD" patients studied in the study were mild based on the H&Y score (2.2 +/- 0.3). Based on published studies, patients with H&Y score about 2 may have bilateral involvement without impairment of balance and posture. It is unclear the rational that authors selected this cohort of PD patient.
· Authors did not reveal whether or not tested patients were under any treatment associated with PD during testing period. Did patients have a drug holiday prior to the test.
· More importantly, ages of 28 healthy controls should be presented/provided since PD is elderly person’s disease (tested patients with mean age of 67 years). Therefore, using age-matched controls should be used for a meaningful comparison. Otherwise, the result generated from the study becomes less important and clinically relevant. It is well documented that Aging is associated with functional deterioration, including in the peripheral sensory structures, thereby affecting vision, hearing, and balance. Additionally, elderly individuals are more likely to suffer from multiple chronic conditions, which often leads to frailty with risk of falls, which presents a serious public health problem. Balance disability increases with age in general [PMC8116987].
Thus, it would be difficult to agree with those conclusions stated by authors without learning aforementioned information.